# Self-Supervised Transformers
# as Iterative Solution Improvers for Constraint Satisfaction

**Yudong W. Xu**[1]  **Wenhao Li**[1]  **Scott Sanner**[1 2]  **Elias B. Khalil**[1 2 3]

## Abstract

We present a Transformer-based framework for Constraint Satisfaction Problems (CSPs). CSPs find use in many applications and thus accelerating their solution with machine learning is of wide interest. Most existing approaches rely on supervised learning from feasible solutions or reinforcement learning, paradigms that require either feasible solutions to these NP-Complete CSPs or large training budgets and a complex expert-designed reward signal. To address these challenges, we propose ConsFormer, a self-supervised framework that leverages a Transformer as a solution refiner. ConsFormer constructs a solution to a CSP iteratively in a process that mimics local search. Instead of using feasible solutions as labeled data, we devise differentiable approximations to the discrete constraints of a CSP to guide model training. Our model is trained to improve random assignments for a single step but is deployed iteratively at test time, circumventing the bottlenecks of supervised and reinforcement learning. Experiments on Sudoku, Graph Coloring, Nurse Rostering, and MAXCUT demonstrate that our method can tackle out-of-distribution CSPs simply through additional iterations.

## 1. Introduction

Constraint Satisfaction Problems (CSPs) are fundamental to many real-world applications such as scheduling, planning, and resource management. However, solving CSPs efficiently in practice remains a significant challenge due to their NP-complete nature. Traditional solvers based on con-

straint propagation and backtracking search can be computationally expensive, especially for large problem instances. This has motivated the exploration of learning-based approaches as fast neural heuristics for CSP solving (Dai et al., 2017; Selsam et al., 2019; Bengio et al., 2021).

Most existing learning-based methods use either supervised or reinforcement learning (RL). Supervised approaches train models on datasets of CSP instances with feasible solutions as labels, a paradigm that is laden with drawbacks. First, generating labels for CSP instances requires solving them, which makes it challenging to generate the large quantities of data often needed to train a model that generalizes well. This is especially true for hard instances that traditional solvers struggle to solve quickly. Second, CSPs often have multiple feasible solutions (Russell & Norvig, 2016), making it difficult to apply supervised learning unambiguously when there are many possible labels for the same input. RL-based methods, on the other hand, search for solution strategies through black-box optimization of a reward function, often requiring extensive computing resources. Designing reward functions that capture solution feasibility across different constraints is difficult yet crucial to success in RL (Arulkumaran et al., 2017). These limitations hinder the generalization and scalability of learned heuristics.

To address these challenges, we introduce ConsFormer, a Transformer-based self-supervised framework for solving CSPs. Inspired by Constraint-based Local Search (Hentenryck & Michel, 2009), ConsFormer learns to iteratively refine variable assignments through a self-supervised training paradigm that approximates discrete constraints with continuous differentiable penalty functions. Our model is trained to improve an initial random assignment in a single refinement step, but is applied iteratively at test time. While a single step may not yield a feasible solution, a sufficiently large number of improving iterations (on average) does. Examples of ConsFormer solutions are shown in Figure 1.

Transformers provide a natural fit for this approach due to their strong generalization capabilities and their ability to process structured data efficiently (Lewkowycz et al., 2022; Achiam et al., 2023). They are particularly effective at learning with tokenized inputs, making them well-suited for combinatorial problems formulated in the Constraint

---

[1]Department of Mechanical & Industrial Engineering, University of Toronto [2]Vector Institute for Artificial Intelligence [3]Scale AI Research Chair in Data-Driven Algorithms for Modern Supply Chains. Correspondence to: Yudong Xu <wil.xu@mail.utoronto.ca>.

*Proceedings of the 42nd International Conference on Machine Learning*, Vancouver, Canada. PMLR 267, 2025. Copyright 2025 by the author(s).

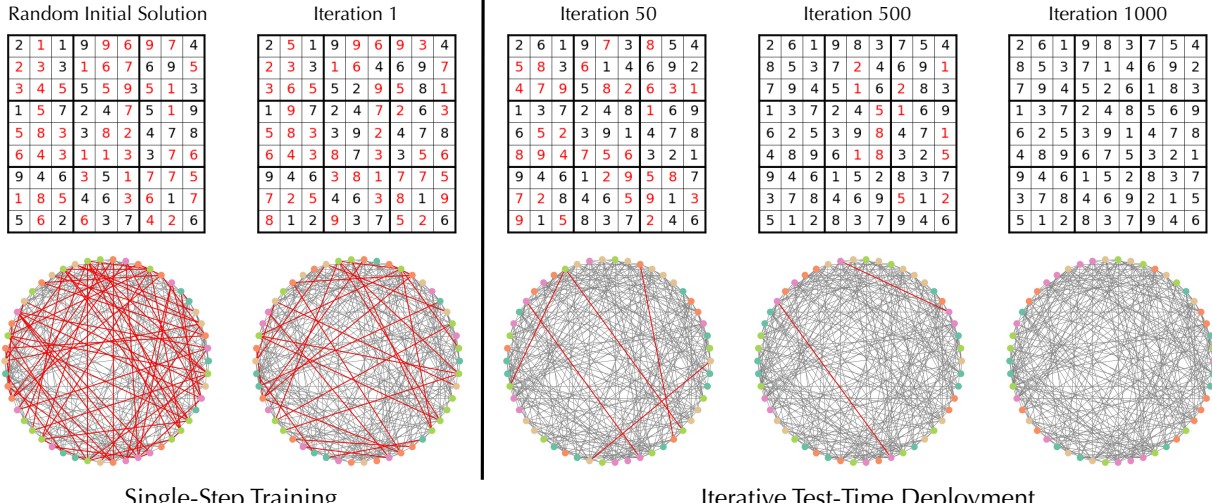

*Figure 1.* ConsFormer models construct solutions for Sudoku (Top) and Graph Coloring (Bottom). The models are trained with a single step from randomly initialized assignment. At test time, a ConsFormer model is invoked iteratively until a feasible solution is found or an iteration limit is met.

Programming paradigm (Rossi et al., 2006). Furthermore, recurrence has been shown to enhance the generalization abilities of Transformers (McLeish et al., 2024; Fan et al., 2025), reinforcing their suitability for our setting.

Our iterative solution improvement strategy enables Cons-Former to generalize beyond its training distribution, effectively solving out-of-distribution (OOD) CSP instances simply by performing more refinement steps. We evaluate ConsFormer on a diverse set of CSP problems, including Sudoku, Graph Coloring, and Nurse Scheduling, demonstrating its ability to generalize across problem domains. Our implementation is available on GitHub[1].

The following high-level findings summarize our work:

– **Self-supervised learning can be applied to solve CSPs.** We show that a loss function that combines differentiable penalties for the violation of the discrete constraints of a CSP can guide model training without the need for labels.

– **Decision variable positional information is key for Transformer learning.** We show that by representing variable positional information as absolute and relational positional encodings in the Transformer, we enable solution improvement in variable space.

– **ConsFormer can generalize when trained to perform solution improvement.** While trained to perform a single improvement step, ConsFormer generalizes to

out-of-distribution instances and achieves state-of-the-art results for generalizing to OOD tasks in Sudoku, outperforming all existing neural methods.

## 2. Background

### 2.1. Constraint Satisfaction and Programming

A Constraint Satisfaction Problem (CSP) is a mathematical model used to represent problems that involve finding values for a set of variables subject to (possibly discrete and non-linear) constraints. Formally, a CSP is defined as a tuple $(X, D, C)$, where $X = \{x_1, x_2, \ldots, x_n\}$ is a finite set of variables, $D = \{D_1, D_2, \ldots, D_n\}, D_i \subset \mathbb{Z} \ \forall i \in [n]$ represents the discrete domains of these variables, and $C = \{c_1, c_2, \ldots, c_m\}$ is the set of constraints, where each constraint $c_i$ is defined over a subset of variables $X_i \subseteq X$, restricting the values that can be simultaneously assigned to them. The goal in solving a CSP is to assign to each variable a value from its domain such that all constraints in $C$ are satisfied.

Constraint Programming (CP) (Rossi et al., 2006) is the study of mathematical models and solution algorithms for CSPs. CP uses highly expressive *global constraints* (Beldiceanu et al., 2005) that involve multiple variables and are designed to capture common constraint structures that appear in a wide range of real-world applications. One prominent example of a global constraint is the ALLDIFFERENT constraint (Régin, 1994), which ensures that a subset of the variables take on distinct values.

[1] https://github.com/khalil-research/ConsFormer

## 2.2. Related Work

**Constraint solving with supervised learning.** Supervised learning has been extensively applied to constraint solving. For example, Pointer Networks (Vinyals et al., 2015) are used for the sequential generation of combinatorial problems involving permutations such as the traveling salesperson problem. Palm et al. (2018) propose a graph-based recurrent network to model CSPs, effectively leveraging the graph structure to refine variable assignments iteratively. SATNet (Wang et al., 2019) differentiates through semidefinite programming relaxations in a supervised setting to handle logical constraints. Du et al. (2024) introduce a method for iterative reasoning through energy diffusion, focusing on progressive refinement of solutions. More recently, Yang et al. (2023) proposed a recurrent Transformer architecture that reuses the same Transformer weights across multiple steps, iteratively refining inputs before projecting them to the final outputs. The common drawback for supervised approaches is the need for labels, which is not easy to generate for many large CSP problems. Additionally, for problems with more than one unique solution, label generation becomes non-trivial.

**Constraint solving without labels.** A common recipe for Reinforcement Learning (RL) in constraint solving is to express the problem with a graph which is then processed using Graph Neural Networks (GNN). The GNN's weights are updated using RL based on a reward function expressing an objective function and/or constraint satisfaction (Dai et al., 2017; Chalumeau et al., 2021; Li et al., 2024; Boisvert et al., 2024). For example, Tönshoff et al. (2023) converts a CSP instance into a tripartite variable-domain-constraint graph which is then solved using a GNN trained by RL. Similarly, Yolcu & Póczos (2019) represent SAT problems using a clause-variable graph. Wu et al. (2022) use a Transformer architecture to learn discrete transformation steps with RL for routing problems. However, RL approaches require significant computational resources for training, as well as an expertly designed reward signal for each problem.

Non-RL based methods require addressing the non-differentiability of discrete constraints. Yang et al. (2022) use the straight-through estimator (Bengio et al., 2013) for logical constraints and Tang et al. (2024) explore a similar approach for mixed-integer non-linear programs. Toenshoff et al. (2021) devise a continuous relaxation for binary constraints (i.e., constraints involving two variables) which are used to guide a recurrent GNN to generate solutions; this approach is limited in applicability as many CSPs of interest have non-binary constraints. Bai et al. (2021) design continuous relaxations for some constraint classes in conjunction with a reconstruction loss to tackle a visual Sudoku problem; it is unclear how their architecture can be adapted to CSP solving in general. Self-supervised learning has been successfully applied in continuous domains (Donti et al.; Park & Van Hentenryck, 2023), as well as for SAT (Ozolins et al., 2022).

**Continuous relaxation of discrete functions.** Continuous relaxations have been used effectively to approximate discrete functions. For example, T-norm has been widely implemented as a continuous approximation for discrete binary logic operations (Petersen et al., 2022; Giannini et al., 2023; Gimelfarb et al., 2024). Petersen et al. (2021) introduced continuous relaxations for discrete algorithms, such as if-statements and while-loops. Combinatorial constraints have also been approximated using entropy-based relaxations (Chen et al., 2019), probabilistic methods (Karalias & Loukas, 2020; Bu et al., 2024), and set function extensions (Karalias et al., 2022).

**Recurrency for generalization.** The incorporation of recurrency has been shown to improve a model's generalization. Bansal et al. (2022) implement recurrent ResNet blocks to solve simple logic puzzles and show that increasing recurrent steps at test-time allows generalization to harder unseen tasks. Recurrency was introduced to the Transformer architecture by sharing weights across Transformer layers (Dehghani et al., 2019; Takase & Kiyono, 2023), yielding improved generalization capabilities on arithmetic and logic-based string manipulation tasks (McLeish et al., 2024; Fan et al., 2025). Our method differs from the existing work as recurrency is only introduced during test-time deployment.

# 3. ConsFormer: a Single-Step Self-Supervised Transformer

We introduce ConsFormer, a single-step Transformer trained with self-supervision. Given an assignment of values to variables (hereafter referred to as *variable assignment*), ConsFormer attempts to generate a refined variable assignment that is closer to satisfying the constraints of the input CSP. An overview of our model is shown in Figure 2.

Section 3.1 presents a Transformer-compatible representation of variable assignments. Section 3.2 details the Transformer design and how it generates an updated assignment. Section 3.3 focuses on the self-supervised training process. Finally, Section 3.4 discusses how the model, trained for single-step solution refinement, can be deployed iteratively at test time to solve CSPs.

## 3.1. Input Representation

The input to the model includes the current variable assignment (which may be infeasible), variable indices, and a binary relational constraint graph indicating the participation of a variable in a constraint. We adapt the Transformer

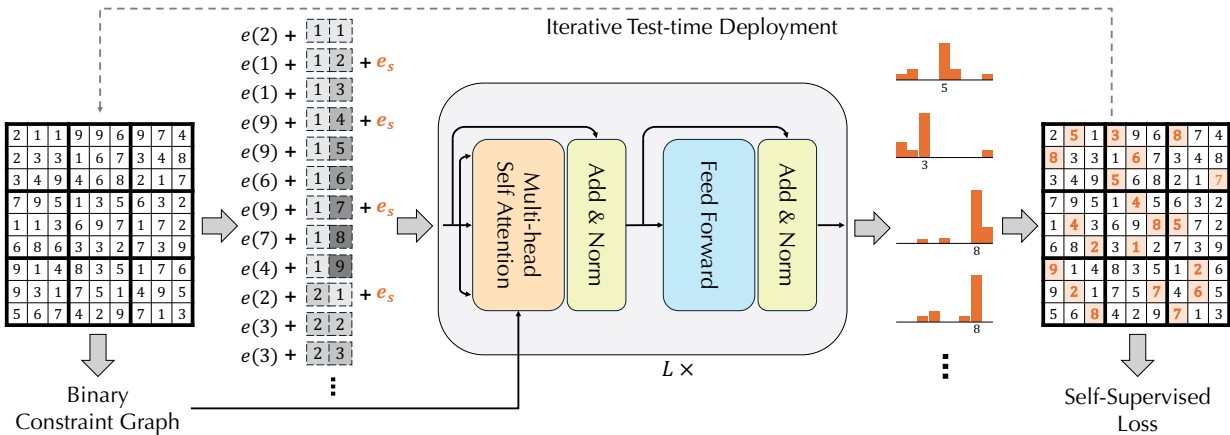

*Figure 2.* ConsFormer: a Single-Step Self-Supervised Transformer for CSP. A CSP instance is transformed into a set of input embeddings composed of the variable assignment value encoding, index-based Absolute Position Encoding, and a selected update set embedding $e_s$ indicating the variables to be updated. The input is processed by $L$ Transformer layers, incorporating the binary constraint graph as Relative Positional Encodings for attention. The output values are used to update the variable assignments which are then used to compute a differentiable self-supervised loss on constraint violation. Although trained to perform one step of solution improvement, ConsFormer can be deployed iteratively at test time, improving the odds of solving instances never seen during training.

architecture to process a CSP instance by encoding these elements as follows.

**Variable assignments as tokens.** Let $X = \{x_1, x_2, \ldots, x_n\}$ be the set of variables in a CSP, each of which has a finite domain $D_i$. A variable assignment $x_i = v$, $v \in D_i$, is treated as a token. A learnable embedding $\mathbf{e}(v)$ is assigned to each unique value $v \in \bigcup_{i=1}^{n} D_i$. The input variable assignment, represented as $\mathbf{X} = \{x_1 = v_1, x_2 = v_2, \ldots, x_n = v_n\}$, forms the input token set to the Transformer. Thus, the input embedding set is given by

$$\mathbf{E} = \{\mathbf{e}(v_1), \mathbf{e}(v_2), \ldots, \mathbf{e}(v_n)\}. \tag{1}$$

In this paper, we focus on problems where all variables share the same domain $D$.

**Representing variable indices with Absolute Positional Encoding.** Transformers use Absolute Positional Encoding (APE) to represent the position of tokens in a sequence. For CSPs, we use APE to encode the indices of variables. If the indices of a variable $x_i$ are multi-dimensional, we concatenate the positional encodings for each dimension. Specifically, let $x_{i_1, i_2, \ldots, i_k}$ denote a variable with $k$-dimensional indices $(i_1, i_2, \ldots, i_k)$. The APE for this variable is computed as:

$$\text{APE}(x_{i_1, i_2, \ldots, i_k}) = \text{Concat}(\text{PE}(i_1), \text{PE}(i_2), \ldots, \text{PE}(i_k)), \tag{2}$$

where $\text{PE}(i_k)$ is the positional encoding for dimension $k$. For example, in Sudoku, a variable $x_{12}$ would have an APE formed by concatenating the encodings for row 1 and column 2. This approach is inspired by the APE design in Vision Transformers (Carion et al., 2020; Li et al., 2025).

**Constraint relations as Relative Positional Encoding.** A Relative Positional Encoding (RPE) is typically used by Transformers to capture the positional relationship between tokens independently of their absolute positions in a sequence. For CSPs, we use RPE to encode the constraint relationships between variables. Specifically, we represent the CSP constraints as a binary constraint graph $\mathbf{G} = (V, E)$, where $V = \{1, 2, \ldots, n\}$ corresponds to the variables and $E$ contains edges between pairs of variables that participate together in at least one constraint of the CSP.

The RPE is incorporated into the attention mechanism by modifying the attention logits. Let $\mathbf{A}_{ij}$ denote the attention logit between variables $i$ and $j$. The modified logits are computed as:

$$\mathbf{A}_{ij} = \mathbf{A}_{ij} + \text{RPE}(i, j),$$

where

$$\text{RPE}(i, j) = c \cdot \mathbb{I}[(i, j) \notin E], \tag{3}$$

and $c \leq 0$ is either a constant hyperparameter or a learned parameter. Setting $c$ to $-\infty$ effectively masks the attention between variables that do not have any constraints in common. The inclusion of the constraint graph via the RPE helps the model identify variable pairs that have a strong effect on each other's assignments.

### 3.2. A Single-Step Transformer Architecture

Our model takes the inputs described in the previous section and outputs a new variable assignment. Below, we outline the key components of the Transformer architecture.

**Variable subset selection.** Inspired by the *local search* principle, we posit that small modifications to a variable assignment are preferable as it is easier to assess their impact on constraint satisfaction. Our model essentially performs a *stochastic* local search by randomly selecting a subset $S \subset X$ of the variables to update. We do this by flipping a biased coin with probability of selection $p$ for each variable, where $p$ is a hyperparameter. A special learned embedding $\mathbf{e}_s$ is added to the variables in $S$. The Transformer's output for the variables in $S$ is used to update the assignment; variables not in $S$ take on the same values as in the input variable assignment. The input to the first Transformer block for a variable $x_i$ is given by:

$$\mathbf{h}_i^{(0)} = \alpha \cdot \mathbf{e}(v_i) + \beta \cdot \text{APE}(x_i) + \gamma \cdot \mathbf{e}_s \cdot \mathbb{I}[x_i \in S], \quad (4)$$

where $\mathbf{e}(v_i)$ is the token embedding of variable $x_i$'s current value $v_i$ as described in Equation (1), and $\text{APE}(x_i)$ is its positional encoding as computed in Equation (2). $\alpha, \beta, \gamma$ are learnable scalars that allow the model to balance the contributions of each encoding, inspired by Li et al. (2025). The set of embeddings for all variables forms the input set $\mathbf{H}^{(0)} = \{\mathbf{h}_1^{(0)}, \mathbf{h}_2^{(0)}, \dots, \mathbf{h}_n^{(0)}\}$.

We note that this allows for easy handling of problems where certain variables have fixed values, such as in Sudoku. We simply bypass the variable subset selection step for the fixed variables, ensuring they are never updated by ConsFormer.

**Self-attention.** ConsFormer employs a multi-head self-attention mechanism to compute updated representations of variables based on other variables. For each variable token $\mathbf{h}_i^{(l)} \in \mathbb{R}^{h \times 1}$ at layer $l$, the self-attention mechanism for a single attention head proceeds as follows:

- Each input token is projected to query, key, and value vectors:

$$\mathbf{q}_i = \mathbf{W}^Q \mathbf{h}_i^{(l)}, \quad \mathbf{k}_i = \mathbf{W}^K \mathbf{h}_i^{(l)}, \quad \mathbf{v}_i = \mathbf{W}^V \mathbf{h}_i^{(l)},$$

where $\mathbf{W}^Q \in \mathbb{R}^{d \times h}$, $\mathbf{W}^K \in \mathbb{R}^{d \times h}$, and $\mathbf{W}^V \in \mathbb{R}^{d_v \times h}$ are learnable weight matrices.

- The relative positional encoding $\text{RPE}(i, j)$ as described in Equation (3) is added to the attention logits $\mathbf{A}_{ij}$:

$$\mathbf{A}_{ij} = \frac{\mathbf{q}_i^\top \mathbf{k}_j}{\sqrt{d}} + \text{RPE}(i, j).$$

- The attention weights are computed using a softmax:

$$\alpha_{ij} = \frac{\exp(\mathbf{A}_{ij})}{\sum_{k \in \mathcal{S}} \exp(\mathbf{A}_{ik})}.$$

- The output representation for token $i$ is computed as:

$$\mathbf{z}_i = \sum_{j \in \mathcal{S}} \alpha_{ij} \mathbf{v}_j.$$

**Feedforward network and layer stacking.** The output of the self-attention mechanism $\mathbf{z}_i$ is passed through a position-wise feedforward network (FFN):

$$\mathbf{h}_i^{(l+1)} = \text{FFN}(\mathbf{z}_i) = \mathbf{W}_2(\text{GeLU}(\mathbf{W}_1 \mathbf{z}_i + \mathbf{b}_1)) + \mathbf{b}_2,$$

where $\mathbf{W}_1$, $\mathbf{W}_2$, $\mathbf{b}_1$, and $\mathbf{b}_2$ are learnable parameters. The Transformer consists of multiple such layers.

**Output: one-hot variable assignments.** At the final layer, the Transformer outputs a one-hot vector over the domain $D_i$ of each variable in the subset $S$, representing its new assignment. Specifically, for variable $x_i$, the output is:

$$\hat{\mathbf{y}}_i = \text{GumbelSoftmax}\left(\mathbf{W}_{\text{out}} \mathbf{h}_i^{(L)} + \mathbf{b}_{\text{out}}\right),$$

where $|\hat{\mathbf{y}}_i| = |D_i|$, $\mathbf{W}_{\text{out}}$ and $\mathbf{b}_{\text{out}}$ are learnable, and $L$ is the number of Transformer layers. The Gumbel-Softmax (Jang et al., 2017) operator serves as a differentiable proxy to selecting the highest-output domain value. The predicted assignment for $x_i$ is then:

$$v_i' = \arg\max \hat{\mathbf{y}}_i, \quad \forall i \in S.$$

### 3.3. Self-supervised Loss Function

How should the Transformer learn to refine an input variable assignment into a better one? In the CSP context, the loss function must reflect the level of constraint satisfaction achieved by the predicted assignment. As argued earlier, one could use a supervised approach in which a feasible solution is first computed for each training CSP and a loss function measuring the variable-wise mismatch between the prediction and the solution is used. However, this approach hinges on solving many NP-Complete CSPs, a substantial overhead for large and complex problems. Additionally, there may be multiple feasible solutions, making supervision by a single solution somewhat arbitrary. Alternatively, our Transformer could be trained by RL, with the reward function reflecting the level of constraint satisfaction. We argue that this is unnecessarily complicated. An input CSP instance is fully observable as is the amount of violation of a constraint for any given variable assignment. Treating this violation signal as part of a reward function given by a black-box "environment" is thus overkill. Should we be able to derive differentiable approximations to the constraints, their violations could be used directly in a loss function, enabling end-to-end differentiable training.

With these design principles in mind, we train our Transformer using self-supervision. As our loss function, we use a linear combination of approximations to the amount of constraint violations by the predicted variable assignment to guide the model towards a satisficing predicted assignment.

*Table 1.* Discrete constraints used in our studied problems and their continuous penalty counterparts. In the continuous penalties, the variables $x_i$ are represented by probability distributions approximating their one-hot form such that $x_i^{(j)} \in [0, 1] \; \forall j \in \{1, \ldots, m\}$, where $\{1, \ldots, m\}$ represents the domain $D_i$. Numerical examples for each constraint can be found in Appendix A.

| **Discrete Constraint** ($c$) | **Continuous Penalty** ($p$) |
|---|---|
| $\text{CARDINALITY}_j(x_1, \ldots, x_n) = k$ | $\left| k - \sum_{i=1}^n x_i^{(j)} \right|$ |

*Explanation:* The cardinality constraint ensures that there are exactly $k$ variables taking the value $j$. The continuous relaxation penalizes the deviation from the desired count $k$.

| | |
|---|---|
| $\text{ALLDIFFERENT}_{m=n}(x_1, \ldots, x_n)$ | $\sum_{j=1}^m \left( \left| 1 - \sum_{i=1}^n x_i^{(j)} \right| \right)$ |

*Explanation:* The all-different constraint ensures that each variable takes a unique value. When the number of variables equals the domain size $m$, the all-different constraint can be rewritten as $n$ cardinality constraints restricting the cardinality of every value in the domain to be 1.

| | |
|---|---|
| $\text{ALLDIFFERENT}_{m>n}(x_1, \ldots, x_n)$ | $\sum_{j=1}^m \left[ \text{ReLU}\left( \sum_{i=1}^n x_i^{(j)} - 1 \right) + \sum_{i=1}^n x_i^{(j)} \cdot \left( \left| 1 - \sum_{i=1}^n x_i^{(j)} \right| \right) \right]$ |

*Explanation:* When there are more domain values than variables, each value should appear no more than once. This is enforced by ensuring values above 1 are penalized and values remain in the set $\{0, 1\}$.

| | |
|---|---|
| $x_i \neq x_k$ | $\sum_{j=1}^m (x_i^{(j)} \cdot x_k^{(j)})$ |

*Explanation:* This constraint ensures that two variables take different values by penalizing overlapping one-hot encodings. The continuous relaxation takes the dot product of the two variables, penalizing it if it is above 0.

However, many constraints in CSP are discrete and not differentiable. To address this, we introduce simple continuous penalties that approximate discrete constraints, which are then used to compute the loss for guiding the model. Let $P = \{p_1, p_2, \ldots, p_m\}$ be the set of continuous penalties approximating constraints $C = \{c_1, c_2, \ldots, c_m\}$ such that

$$p_i(X_i) = 0 \iff c_i(X_i) = \texttt{True},$$

implying that $X^*$ is a feasible solution for the CSP when

$$p_i(X_i^*) = 0 \quad \forall p_i \in P.$$

The loss for ConsFormer for a single CSP training instance is therefore

$$\mathcal{L} = \sum_i \lambda_i f(p_i(X_i)) \quad \forall p_i \in P, \tag{5}$$

where hyperparameter $\lambda_i$ is the weight assigned to $p_i$, and $f$ is an optional operation to transform the penalty for better learning. In practice, we implemented the common quadratic penalty, $f(x) = x^2$. The discrete constraints and their relaxed continuous counterparts we implemented for our experiments are shown in Table 1. Further discussion about the design process can be found in Appendix H and numerical examples of valid and invalid assignments for each constraint can be found in Appendix A.

### 3.4. Iterative Test-Time Deployment

Another issue with RL is its multi-step nature which requires exploring an extremely large space of iterative solution refinement sequences. We show that learning a single-step solution refiner with self-supervision suffices as the model can be be deployed iteratively at test time. More specifically, our method refines an initial solution by repeatedly feeding its output variable assignment back as input to the next iteration as visualized in Figure 2.

In this sense, ConsFormer can be viewed as performing a single step of neural local search to improve the candidate solution. Our experiments focus on basic iterative solution refinement in one continuous sequence, without additional augmentations. However, this capability can be combined with techniques such as backtracking and random restarts (Rossi et al., 2006) to create a neuro-symbolic solver. Another possible extension is to incorporate ConsFormer as an evolutionary algorithm (Holland, 1992) utilizing the Transformer's parallel processing ability to update a pool of candidate solutions all at once. While we leave these explorations as future work, we demonstrate the potential of this direction by implementing a simple multi-start strategy, which we show can significantly enhance performance in Appendix F.

# 4. Experimental Results

## 4.1. Problems

**Sudoku** is a well-known CSP problem that involves filling a $9 \times 9$ grid with digits from 1 to 9 such that each row, column, and $3 \times 3$ sub-grid contain each digit exactly once. A single Sudoku instance consists of a partially filled board and a unique assignment to the unfilled cells that satisfies the constraints. A Sudoku instance's difficulty is determined by the initial board: fewer initially filled cells in the board involve a larger space of possible assignments to the unfilled cells, with the hardest Sudoku puzzles having only 17 of the 81 numbers provided (McGuire et al., 2014). We use the ALLDIFFERENT$_{m=n}(x_1, \ldots, x_n)$ constraint from Table 1 to formulate the problem and its corresponding continuous penalty as the loss function to guide the model learning. The full formulation of Sudoku in CP is detailed in Appendix B.

We use the dataset from SATNet (Wang et al., 2019), which contains instances with $[31, 42]$ missing values, as the training and in-distribution testing dataset. To test our model's ability to generalize to harder out-of-distribution instances, we use the dataset from RRN (Palm et al., 2018) which contains instances with $[17, 34]$ missing values.

**Graph Coloring** is the problem of finding an assignment of colors to vertices in a graph such that no two neighboring nodes share the same color. The problem is defined by the graph's structure and the number of available colors $k$. We generate two sets of graph instances for $k = 5$ and $k = 10$ following a similar procedure as Tönshoff et al. (2023) (See Appendix C for details). Training graphs have 50 vertices for $k = 5$ and 100 vertices for $k = 10$ whereas OOD graphs have 100 for $k = 5$ and 200 for $k = 10$. We use inequality constraints of the form $x_i \neq x_j$ for an edge between nodes $i$ and $j$ and their penalty in Table 1 to represent the coloring constraints.

**Nurse Scheduling** is an operations research problem of assigning nurses to shifts. A problem instance has a specified number of days $n$, number of shifts per day $s$, number of nurses per shift $ns$, and a total number of nurses. The variables $x_{d,n,ns}$ are the shift slots indexed by the day, shift, and nurse and their domains are the indices of the nurses. A feasible solution ensures that no nurse is assigned to more than one shift per day and avoids assigning the same nurse to both the last shift of one day and the first shift of the following day. We use both the inequality $x_i \neq x_j$ and the ALLDIFFERENT$_{m>n}(x_1, \ldots, x_n)$ constraints for this problem; see Appendix B for the full formulation.

**MAXCUT** aims to partition the vertices of a graph into two disjoint sets such that the number of edges crossing the partition is maximized. The problem can be viewed as a 2-coloring problem where the objective is to satisfy as many inequality constraints $x_i \neq x_j$ as possible. Following the

*Table 2.* Performance comparison for Sudoku. In-distribution test instances contain $1,000$ instances from the SATNet dataset, OOD refers to Out-of-Distribution evaluation on the RRN test dataset which contains 18K instances. *Values reported in (Du et al., 2024).

| Method | Test Instances | Harder OOD Instances |
|---|---|---|
| Wang et al. (2019) * | 98.3 | 3.2 |
| Palm et al. (2018)* | 99.8 | 28.6 |
| Yang et al. (2023) | **100** | 32.9 |
| Yang et al. (2023) (2k Iters) | 97.7 | 14.0 |
| Du et al. (2024) * | 99.4 | 62.1 |
| ConsFormer (2k Iters) | **100** | 65.88 |
| ConsFormer (10k Iters) | **100** | **77.74** |

same setup as Tönshoff et al. (2023), we generate random graphs with 100 vertices for training and evaluate generalization on benchmark instances from the GSET dataset (Ye, 2003), which includes weighted graphs with sizes ranging from 800 to 10000 vertices.

## 4.2. Training

For each of the problems, we train the model with randomly initialized variable assignments guided by the loss function defined in Equation (5) and the corresponding $p_i$ associated with the constraints used to define the CSP. The training set contains 9K instances for all problems. The training details and hyperparameters for the best performing model for each problem is detailed in Appendix D.

## 4.3. Results

**Sudoku.** Table 2 reports the performance of ConsFormer and various neural methods on the Sudoku task. ConsFormer solves 100% of the Sudoku tasks from the in-distribution test dataset. On the harder out-of-distribution dataset, ConsFormer significantly outperforms all learned methods, demonstrating superior generalization capabilities. ConsFormer achieves instance solve rates of 65.88% and 77.74% with 2k and 10k iterations, respectively.

This highlights the iterative reasoning nature of our approach. Harder instances can be solved with additional reasoning steps, whereas other solvers with fixed reasoning steps struggle. Notably, Yang et al. (2023)'s approach also employs iterative reasoning with Transformers, but their performance degraded with more test-time iterations while ConsFormer's continued to improve. This could be due to Yang et al. (2023)'s approach being trained for 32 iterations, while ConsFormer was trained for a single iteration, allowing it to generalize better when applied iteratively.

*Table 3.* Performance comparison for Graph-Coloring tasks. OOD refers to Out-of-Distribution evaluation for ANYCSP and Cons-Former where the number of verticies $n$ in the graph is larger than that of the training instances. All datasets has 1200 instances.

| Method | Test Instances | Harder OOD Instances |
|---|---|---|
| **Graph-Coloring-5** ($n = 50 \rightarrow n = 100$) | | |
| OR-Tools (10s) | **83.08** | **57.16** |
| ANYCSP (10s) | 79.17 | 34.83 |
| ConsFormer (10s) | 81.00 | 47.33 |
| **Graph-Coloring-10** ($n = 100 \rightarrow n = 200$) | | |
| OR-Tools (10s) | 52.41 | 10.25 |
| ANYCSP (10s) | 0.00 | 0.00 |
| ConsFormer (10s) | **52.60** | **11.92** |

*Table 4.* Performance comparison for MAXCUT tasks on GSET. Numbers reported are the average percentage gap to the best known cut size, the lower the better.

| Method | $|V|=800$ | $|V|=1K$ | $|V|=2K$ | $|V|\geq 3K$ |
|---|---|---|---|---|
| Greedy | 5.26 | 6.64 | 6.81 | 6.30 |
| SDP | 3.14 | 4.24 | - | - |
| RUNCSP | 2.38 | 2.90 | 3.30 | 3.26 |
| ECO-DQN | 0.83 | 1.01 | 1.45 | 3.49 |
| ECORD | 0.11 | 0.16 | 0.36 | 1.53 |
| ANYCSP | **0.02** | **0.05** | **0.12** | **0.42** |
| ConsFormer | 0.31 | 0.34 | 0.43 | 1.27 |
| OR-Tools | 1.84 | 2.09 | 3.38 | 3.08 |

**Graph Coloring.** Table 3 summarizes the performance of OR-Tools, ANYCSP (Tönshoff et al., 2023), and Cons-Former on Graph Coloring instances. OR-Tools is a state-of-the-art traditional solver for constraint programming applications and serves as a strong baseline (Perron & Didier), achieving 100% on Sudoku instances. We ran test instances sequentially through all models with a 10-second timeout, though ConsFormer can process instances significantly faster if processed in batches due to the Transformer architecture. We note that the harder dataset is not out of distribution for OR-Tools, since it solves each task individually and is not a learning-based solver.

ConsFormer demonstrates competitive performance on in-distribution Graph Coloring with $k = 5$, solving 81% of the test instances approaching the 83.08% achieved by OR-Tools. While it lags behind the state-of-the-art solver with 47.33% on the harder OOD test set, ConsFormer outperforms ANYCSP on both distributions, which again shows our method's high generalization ability.

On the more challenging Graph Coloring with $k = 10$, ConsFormer surpasses OR-Tools in performance, showcasing the advantage of learned heuristic approaches: it may not surpass state-of-the-art solvers on smaller instances, but it excels in complex cases under short time limits—crucial for real-world applications. Surprisingly, ANYCSP failed to solve any instances within 10 seconds for both datasets, underscoring the scalability limitations of its graph-based representation and the difficulty of training with RL. Some additional baselines can be found in Appendix G.

**Nurse Scheduling.** For the Nurse Scheduling problem, ConsFormer matches OR-Tools in solving 100% of tasks across both in-distribution and out-of-distribution instances within the 10-second timeout. This high accuracy is expected, given the large number of feasible solutions for each instance, as detailed in Appendix C. However, solving this

problem neurally is non-trivial, as the model must learn to balance multiple constraints within a single step of solution refinement. ANYCSP was not evaluated on this dataset due to their difficulty dealing with the ALLDIFFERENT constraint. This highlights ConsFormer's potential to generalize to complex problems with diverse constraint structures.

**MAXCUT.** Table 4 compares various methods on GSET instances, reported as the relative gap (in percentage) to the best known cut values (Matsuda, 2019). ConsFormer and OR-Tools performances are obtained using the same setup as ANYCSP, while the rest are computed directly from values reported by Tönshoff et al. (2023). While ANYCSP remains the best-performing method on GSET, ConsFormer achieves an average relative gap of 0.31% to 1.27% without extensive model tuning, demonstrating its ability to scale to larger problems with thousands of constraints.

### 4.4. Ablations

**Effect of subset improvement.** Figure 3 examines the impact of varying probability of selection $p$ on the performance of the model. The horizontal axis refers to the number of iterations at test time while the vertical axis represents the percentage of in-distribution test instances solved. We investigate the behavior of the model under different probabilities $p \in \{1.0, 0.9, 0.7, 0.5, 0.3, 0.1\}$, where $p$ determines the probability of selecting each variable for updates during a single iteration as detailed in Section 3.2. A larger $p$ results in more variables being selected to be updated.

When $p = 1.0$ (blue line), all variables are selected for updates during every iteration. This approach leads to rapid improvement in the early stages, as the model converges quickly to local optima. This is clearly observed in Graph Coloring, where the $p = 1.0$ model rapidly solved 65% of instances. Performance plateaus after the initial surge whereas the stochastic models surpass it in accuracy after 50 iterations. The difference in model performance is even more drastic for Sudoku, with the $p = 1.0$ model reaching 20% instances solved early and converging, while the $p =$

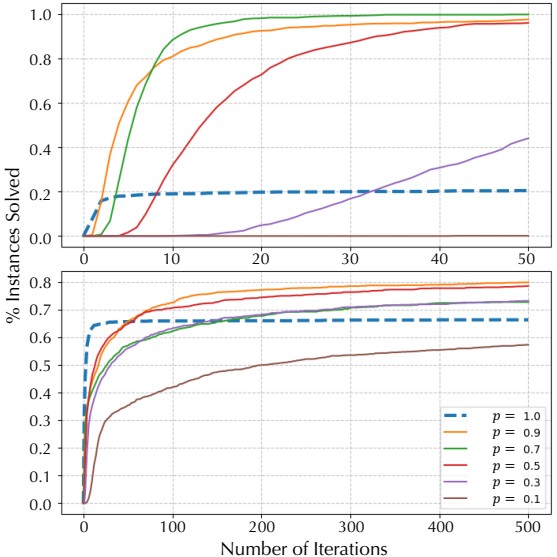

*Figure 3.* Variable selection probability ablation for Sudoku (Top) and Graph Coloring with $k = 5$ (Bottom). The horizontal axis shows the number of iterations at test time and the vertical axis represents the % solved of in-distribution test instances.

0.9, $p = 0.7$, and $p = 0.5$ models surpass it and reach near 100% within 50 iterations.

These findings highlight the importance of incorporating stochasticity into the update process for combinatorial optimization tasks. While deterministic updates provide faster initial convergence, they are prone to premature stagnation. Stochastic updates, by selectively updating a subset of variables, improve generalization and allow the model to achieve higher final performance.

**Effect of variable information as positional encodings.** Table 5 and 6 show the performance of ConsFormer with different positional encodings. The value displayed indicates the percentage of in-distribution test instances solved by the model running $1,000$ iterations

Across both Graph Coloring and Sudoku, we observe that the inclusion of relative variable relations with RPE provides a significant performance boost. This is especially true in the Graph Coloring problem, which heavily relies on the constraint graph, since the vertices have no inherent ordering to them, and therefore the indices have little meaning for the model to learn from.

In Sudoku however, we see that the Transformer is able to achieve strong performance using only 2D APE, without leveraging explicit constraint graph information. This indicates that in highly structured problems like Sudoku, the positional indices of variables alone contain sufficient information for solving the instances. We also observe that

*Table 5.* Positional Encoding Ablation on Sudoku.

| Model Variant | No APE | 1D APE | 2D APE |
|---|---|---|---|
| **No RPE** | 0.00% | 0.00% | 99.90% |
| **Learned RPE** | 98.70% | 97.10% | 98.20% |
| **Masked RPE** | 99.80% | 99.50% | 99.80% |

*Table 6.* Positional Encoding Ablation on Graph Coloring.

| Model Variant | No APE | | 1D APE | |
|---|---|---|---|---|
| | COL50 | COL100 | COL50 | COL100 |
| **No RPE** | 1.58% | 0.00% | 1.25% | 0.00% |
| **Learned RPE** | 78.00% | 12.50% | 77.33% | 0.25% |
| **Masked RPE** | 75.67% | 52.25% | 77.00% | 51.92% |

2D APE outperforms the standard 1D APE typically used in Transformers. These results suggests the importance of supporting both forms of positional encodings, as different properties of different problems requires distinct spatial or relational information.

We also note that when RPE is implemented as masked RPE, attention scores for each variable are restricted to its connected variables, closely resembling the behavior of a graph neural network with attention.

## 5. Limitations and Future Work

In its current form, ConsFormer assumes a fixed constraint structure, the effects of which on solution feasibility are implicitly learned during training via the loss function. ConsFormer does not receive explicit constraint representations as input. Constraints which are "parametric", e.g., a SAT clause in which some variables may be negated and some not, cannot be handled with the architecture described herein. It is possible to address this limitation through explicit constraint representations in the input; this is an important direction for future work.

Other future work includes exploring neural-symbolic approaches incorporating ConsFormer such as those discussed in Section 3.4, other performance boosting techniques such as self-improvement to augment the training data (Lee et al., 2025), as well as extending ConsFormer to more problems and more constraints, with the goal of devising a general continuous approximation for constraints.

## 6. Conclusion

We introduced ConsFormer, a self-supervised Transformer for iteratively solving Constraint Satisfaction Problems. We showed that our method, trained to perform a single step of solution improvement, is able to generalize to harder out-of-distribution instances at test time, outperforming supervised and reinforcement learning approaches.

## Acknowledgments

We thank the anonymous reviewers for their insightful feedback, which helped us identify key limitations and promising directions for future work. This work was supported by the Institute of Information & Communications Technology Planning & Evaluation (IITP) grant funded by the Korean Government (MSIT) (No. RS-2024-00457882, National AI Research Lab Project). Elias B. Khalil acknowledges support from the SCALE AI Research Chair program.

## Impact Statement

This paper presents work whose goal is to advance the field of Machine Learning. There are many potential societal consequences of our work, none which we feel must be specifically highlighted here.

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

# A. Example Continuous Penalty Evaluations

*Table 7.* Example assignments illustrating the evaluation of continuous penalties for discrete constraints. The variables $x_i$ are represented by probability distributions approximating their one-hot form. Two assignments are shown: a valid assignment representing a set of variable values that exactly satisfy the constraint, and an invalid assignment representing a set of variable values that violate the constraint. The penalty evaluates to 0 for valid assignments and increases with the degree of constraint violation.

| Constraint and Relaxation | Example Assignments |
|---|---|
| $\text{CARDINALITY}_j(x_1, \ldots, x_n) = k$ 

 $\implies \quad p = \left| k - \sum_i x_i^{(j)} \right|.$ 

 $j = 1, k = 2$ | **Valid Assignment:** $x_1 = (1, 0), \ x_2 = (1, 0), \ x_3 = (0, 1)$ 
 $\sum_i x_i^{(1)} = 1 + 1 + 0 = 2$ 
 $p = |2 - 2| = 0.$ 

 **Invalid Assignment:** $x_1 = (0.7, 0.3), \quad x_2 = (0.2, 0.8), \quad x_3 = (0, 1)$ 
 $\sum_i x_i^{(1)} = 0.7 + 0.2 + 0 = 0.9$ 
 $p = |2 - 0.9| = 1.1.$ |
| $\text{ALLDIFFERENT}_{m=n}(x_1, \ldots, x_n)$ 

 $\implies p = \sum_j \left| 1 - \sum_i x_i^{(j)} \right|.$ | **Valid Assignment:** $x_1 = (1, 0, 0), \ x_2 = (0, 1, 0), \ x_3 = (0, 0, 1)$ 
 $\sum_i x_i^{(1)} = 1, \ \sum_i x_i^{(2)} = 1, \ \sum_i x_i^{(3)} = 1$ 
 $p = |1 - 1| + |1 - 1| + |1 - 1| = 0.$ 

 **Invalid Assignment:** $x_1 = (0.9, 0.1, 0), \ x_2 = (0.9, 0.1, 0), \ x_3 = (0, 0, 1)$ 
 $\sum_i x_i^{(1)} = 1.8, \ \sum_i x_i^{(2)} = 0.2, \ \sum_i x_i^{(3)} = 1$ 
 $p = |1 - 1.8| + |1 - 0.2| + |1 - 1| = 0.8 + 0.8 + 0 = 1.6.$ |
| $\text{ALLDIFFERENT}_{m>n}(x_1, \ldots, x_n)$ 

 $\implies p = \sum_j \left[ \text{ReLU}\left( \sum_i x_i^{(j)} - 1 \right) \right.$ 

 $\left. + \sum_i x_i^{(j)} \left| 1 - \sum_i x_i^{(j)} \right| \right].$ | **Valid Assignment:** $x_1 = (1, 0, 0), \ x_2 = (0, 1, 0)$ 
 $\sum_i x_i^{(1)} = 1, \ \sum_i x_i^{(2)} = 1, \ \sum_i x_i^{(3)} = 0$ 
 $\text{ReLU}(1 - 1) = 0, \ 1 \cdot |1 - 1| = 0$ 
 $\text{ReLU}(1 - 0) = 0, \ 0 \cdot |1 - 0| = 0$ 
 $p = 0 + 0 + 0 + 0 + 0 + 0 = 0.$ 
 **Invalid Assignment:** $x_1 = (0.6, 0.4, 0), \quad x_2 = (0.7, 0.3, 0)$ 
 $\sum_i x_i^{(1)} = 1.3, \ \sum_i x_i^{(2)} = 0.7, \ \sum_i x_i^{(3)} = 0$ 
 $\text{ReLU}(1.3 - 1) = 0.3, \ 1.3 \cdot |1 - 1.3| = 0.39$ 
 $\text{ReLU}(0.7 - 1) = 0, \ 0.7 \cdot |1 - 0.7| = 0.21$ 
 $p = 0.3 + 0.39 + 0 + 0.21 + 0 + 0 = 0.9.$ |
| $x_i \neq x_k$ 

 $\implies \quad p = \sum_j \left( x_i^{(j)} x_k^{(j)} \right).$ | **Valid Assignment:** $x_i = (1, 0, 0), \ x_k = (0, 1, 0)$ 
 $p = (1 \cdot 0) + (0 \cdot 1) + (0 \cdot 0) = 0.$ 

 **Invalid Assignment:** $x_i = (0.7, 0.3, 0), \quad x_k = (0.7, 0.2, 0.1)$ 
 $p = (0.7 \cdot 0.7) + (0.3 \cdot 0.2) + (0 \cdot 0.1) = 0.49 + 0.06 + 0 = 0.55.$ |

# B. Constraint Programming formulations

### B.1. Sudoku

We define the Sudoku problem as a constraint satisfaction problem (CSP) with the following components:

**Variables:** Let $X_{i,j}$ denote the variable representing the value assigned to cell $(i,j)$, where $i, j \in \{1, 2, \ldots, 9\}$.

**Domains:** Each variable $X_{i,j}$ takes values from the discrete domain:

$$X_{i,j} \in \{1, 2, \ldots, 9\}.$$

**Constraints:** The solution must satisfy the following AllDifferent constraints:

- Each row must contain unique values:

$$\text{ALLDIFFERENT}_{m=n}(X_{i,1}, X_{i,2}, \ldots, X_{i,9}), \quad \forall i \in \{1, \ldots, 9\}.$$

- Each column must contain unique values:

$$\text{ALLDIFFERENT}_{m=n}(X_{1,j}, X_{2,j}, \ldots, X_{9,j}), \quad \forall j \in \{1, \ldots, 9\}.$$

- Each $3 \times 3$ subgrid must contain unique values. Let $(r, c)$ index the subgrid with $r, c \in \{0, 1, 2\}$, then:

$$\text{ALLDIFFERENT}_{m=n} \begin{pmatrix} X_{3r+1,3c+1}, X_{3r+1,3c+2}, X_{3r+1,3c+3}, \\ X_{3r+2,3c+1}, X_{3r+2,3c+2}, X_{3r+2,3c+3}, \\ X_{3r+3,3c+1}, X_{3r+3,3c+2}, X_{3r+3,3c+3} \end{pmatrix}, \quad \forall r, c \in \{0, 1, 2\}.$$

### B.2. Graph Coloring

Given a graph $G = (V, E)$, we define the graph coloring problem as a constraint satisfaction problem (CSP) with the following components:

**Variables:** Let $X_v$ be a variable representing the color assigned to vertex $v \in V$.

**Domains:** Each variable $X_v$ takes values from a set of $k$ available colors:

$$X_v \in \{1, 2, \ldots, k\}, \quad \forall v \in V.$$

**Constraints:** The solution must satisfy that any two adjacent vertices must be assigned different colors:

$$X_u \neq X_v, \quad \forall (u, v) \in E.$$

### B.3. Nurse Rostering

We define the nurse rostering problem as a constraint satisfaction problem (CSP) with the following components:

**Variables:** Let $x_{d,s,ns}$ be a variable representing the nurse assigned to the $ns$-th slot of shift $s$ on day $d$, where:

$$x_{d,s,ns} \in \{1, 2, \ldots, N\}, \quad \forall d \in \{1, \ldots, n\}, \quad \forall s \in \{1, \ldots, S\}, \quad \forall ns \in \{1, \ldots, NS\}.$$

**Constraints:** A feasible schedule must satisfy the following constraints:

- No nurse can be assigned to more than one shift per day:

$$\text{ALLDIFFERENT}_{m>n}(x_{d,1,1}, x_{d,1,2}, \ldots, x_{d,1,NS}, x_{d,2,1}, x_{d,2,2}, \ldots, x_{d,2,NS}, \ldots, x_{d,S,1}, x_{d,S,2}, \ldots, x_{d,S,NS})$$
$$\forall d \in \{1, \ldots, n\}.$$

- A nurse cannot be assigned both the last shift of a given day and the first shift of the following day:

$$x_{d,S,ns} \neq x_{d+1,1,ns'}, \quad \forall d \in \{1, \ldots, n-1\}, \quad \forall ns, ns' \in \{1, \ldots, NS\}.$$

# C. Dataset Details

## C.1. Graph Coloring

Following (Tönshoff et al., 2023), we generate Graph Coloring instances with the following 3 distributions:

- **Erdős-Rényi graphs** with edge probability $p \sim U[0.1, 0.3]$

- **Barabási-Albert graphs** with parameter $m \sim U[2, 10]$

- **Random geometric graphs** with vertices distributed uniformly at random in a 2-dimensional $1 \times 1$ square and edge threshold radius drawn uniformly from $r \sim U[0.15, 0.3]$.

The 5-coloring instances were drawn uniformly for all 3 distributions, with vertices count 50 for training and in-distribution testing data, vertices count 100 for out of distribution testing. The 10-coloring instances were drawn uniforming from Erdős-Rényi graphs and Random geometric graphs, with vertices count 100 for training and in-distribution testing data, and 200 for out of distribution testing.

For each graph $G$ generated a linear time greedy coloring heuristic as implemented by `NetworkX` (Hagberg et al., 2008) to color the graph without conflict. If the greedy heuristic required $k'$ colors for $G$, then we pose the problem of coloring $G$ with $k$ colors as the training CSP instance, where $k$ is chosen as:

$$k = \max\{3, \min\{10, k' - 1\}\}$$

We generate instances until a fixed number of instances for a specific k is reached. 9000 for training sets, 1200 for test sets.

## C.2. Nurse Scheduling

We generate Nurse Rostering instances with varying difficulties. Each problem instance is defined by the number of days $n$, number of shifts per day $s$, number of nurses required per shift $ns$, and the total number of available nurses $N$.

- **in-distribution instances** were generated with $n = 10$ days, $s = 3$ shifts per day, $ns = 3$ nurses per shift, and a total of $N = 10$ nurses.

- **Out-of-distribution instances** were generated with $n = 10$, $s = 3$, $ns = 3$, and $N = 10$.

The in-distribution instances consisted of 9000 training instances and 1000 test instances. Out-of-distribution instances also had 1000 samples.

To initialize different instances, we assign one random shift to every nurse as an initial assignment. This ensures that each instance starts with a minimally constrained but valid configuration.

We note that these instances are relatively easy to solve due to the large number of feasible solutions available. The constraints in the problem formulation do not drastically limit the space ofValid Assignments. The purpose of this dataset is to examine ConsFormer's ability to solve instances with a combination of different constraints.

## D. Model Details

Our models were trained on various single-core GPU nodes, including P100, V100, and T4. A grid search was conducted to determine the best-performing hyperparameters, evaluating a few hundred configurations per problem. The final reported models were trained with a batch size of 512 for 5000 epochs. The typical training time for a model ranges from 6 to 10 hours (wall clock). The hyperparameters for the best performing model for each of the problems is shown in Table 8. For all models, we used AdamW as the optimizer and applied a dropout of 0.1, with learning rate set to 0.0001.

*Table 8.* Hyperparameters for the best performing models.

|  | Sudoku | Graph-coloring-5 | Graph-coloring-10 | Nurse Scheduling | MAXCUT |
|---|---|---|---|---|---|
| Layer Count | 7 | 4 | 7 | 7 | 4 |
| Head Count | 3 | 3 | 3 | 3 | 3 |
| Embedding Size | 128 | 128 | 128 | 126 | 128 |
| Selection Probability $p$ | 0.5 | 0.3 | 0.3 | 0.3 | 1.0 |

## E. Effects of Gumbel Softmax

We use Gumbel-Softmax in ConsFormer to enable differentiable sampling of discrete variables, aligning with the discrete nature of CSPs.

To better understand why Gumbel-Softmax improves performance, we investigate whether the benefit stems from the stochasticity introduced by Gumbel noise or simply from producing sharper output distributions. To isolate these factors, we compare against a softmax variant with temperature control:

$$\text{Softmax}_\tau(z_i) = \frac{\exp(z_i/\tau)}{\sum_j \exp(z_j/\tau)}$$

This variant allows us to control the sharpness of the output distribution without introducing stochasticity. Table 9 presents results for Sudoku and Graph Coloring (percentage of instances solved), and MAXCUT (absolute gap to best known values).

Results indicate that softmax with temperature performs competitively or even better on smaller-scale problems such as Sudoku and Graph Coloring instances. However, for larger problems like MAXCUT with thousands of variables, Gumbel-Softmax clearly outperforms temperature-controlled softmax.

This suggests that while sharper distributions (e.g., from low temperature) are beneficial, the stochasticity allows the model to generalize better across larger problem instances. The randomness can promote diversity in intermediate solutions which may help the model escape local optima over multiple inference steps.

*Table 9.* Comparison of ConsFormer with and without Gumbel-Softmax

| Method | Gumbel-Softmax | | Softmax | |
|---|---|---|---|---|
| | $\tau = 0.1$ | $\tau = 1$ | $\tau = 0.1$ | $\tau = 1$ |
| Sudoku | 100 | 100 | 100 | 100 |
| Sudoku OOD | 77.74 | 83.71 | **85.67** | 73.72 |
| Graph-Coloring-5 V=50 | **78.16** | 76.91 | 77.33 | 74.91 |
| Graph-Coloring-5 V=100 | 42.50 | 41.08 | **42.66** | 35.33 |
| Graph-Coloring-10 V=100 | 52.60 | 53.25 | **53.66** | 53.0 |
| Graph-Coloring-10 V=200 | 11.92 | 12.75 | **12.92** | 12.75 |
| MAXCUT $|V|=800$ | **24.44** | 102.89 | 123.11 | 126.56 |
| MAXCUT $|V|=1K$ | **18.22** | 44.0 | 58.33 | 56.89 |
| MAXCUT $|V|=2K$ | **47.0** | 119.33 | 123.67 | 135.11 |
| MAXCUT $|V|\geq 3K$ | **155.88** | 187.0 | 287.38 | 305.25 |

# F. ConsFormer with Multi-Start

A simple extension to enhance our model is the implementation of multi-start. Instead of a single initial solution continuously refined, we maintain a pool of candidate solutions that are updated concurrently. A solution is accepted as soon as any candidate satisfies all constraints. This approach naturally complements ConsFormer, as the Transformer architecture efficiently handles batched processing.

Table 10 reports results for various candidate pool sizes. A candidate count of 1 corresponds to the standard version of ConsFormer used in the main paper.

We observe that as the number of candidates increases, the number of update iterations per candidate slightly decreases due to the fixed time budget. However, the drop is modest compared to the increase in candidates, highlighting the scalability of the Transformer-based model. We observe that the multi-start strategy is able to noticably improve model performance, boosting accuracy from 47.33% to 55.67% for graph coloring with $k = 5$ and 11.92% to 15.00% for $k = 10$.

These findings highlight the potential of integrating with symbolic strategies, such as restarts and backtracking, as discussed in Section 3.4. We leave further exploration of these hybrid techniques to future work.

*Table 10.* Performance comparison for Graph-Coloring tasks on Out-of-Distribution evaluation for ConsFormer. Candidates Count refers to the number of solutions used for multi-start. # Iterations Average shows the number of iterations each candidate went through under the time limit.

| Method | Harder OOD Instances | Pool Size | # Iterations Avg |
|---|---|---|---|
| **Graph-Coloring-5** ($n = 100$) | | | |
| OR-Tools (10s) | **57.16** | - | - |
| ConsFormer (10s) | *47.33* | 1 | 2310 |
| ConsFormer (10s) | 43.42 | 2 | 2213 |
| ConsFormer (10s) | 50.92 | 10 | 1634 |
| ConsFormer (10s) | 55.17 | 50 | 1613 |
| ConsFormer (10s) | 55.67 | 100 | 892 |
| **Graph-Coloring-10** ($n = 200$) | | | |
| OR-Tools (10s) | 10.25 | - | - |
| ConsFormer (10s) | *11.92* | 1 | 1490 |
| ConsFormer (10s) | 13.67 | 2 | 1445 |
| ConsFormer (10s) | 13.92 | 5 | 1064 |
| ConsFormer (10s) | 14.42 | 10 | 1150 |
| ConsFormer (10s) | **15.00** | 50 | 806 |
| ConsFormer (10s) | 13.25 | 100 | 229 |

# G. Additional Baselines for Graph Coloring

We provide additional baselines for comparison on the Graph Coloring task. We first run OR-Tools for additional time (30 and 60 seconds) with 10 colors (where ConsFormer had previously outperformed it under 10s). We see that CP-SAT can outperform our method on small instances (nodes=100) with extended time, but it still underperforms on larger instances (nodes=200), even with 6x more time. Furthermore, in the new MAXCUT problem, 20 parallel runs–each with 180s limit–were used to compute the results. ConsFormer also outperforms OR-Tools by a significant margin.

We then include 3 additional heuristic baselines: Greedy Coloring, Feasibility Jump, Random Search. The first is the greedy coloring algorithm implemented by networkx and the other two are local search approaches implemented by OR-Tools. Results are shown in Table 11. We observe that while the local-search based heuristics were able to perform well on the smaller instances, their performance significantly worsens on the larger instances with 10 colors.

*Table 11.* Performance comparison for Graph-Coloring tasks. OOD refers to Out-of-Distribution evaluation for ANYCSP and ConsFormer where the number of verticies $n$ in the graph is larger than that of the training instances. All datasets has 1200 instances.

| Method | Test Instances | Harder OOD Instances |
|---|---|---|
| **Graph-Coloring-5** ($n = 50 \rightarrow n = 100$) | | |
| Greedy | 32.42 | 0.0 |
| OR-Tools-FJ (10s) | 82.83 | 54.5 |
| OR-Tools-RS (10s) | 83.08 | 56.91 |
| OR-Tools (10s) | **83.08** | **57.16** |
| ANYCSP (10s) | 79.17 | 34.83 |
| ConsFormer (10s) | 81.00 | 47.33 |
| **Graph-Coloring-10** ($n = 100 \rightarrow n = 200$) | | |
| Greedy | 0.75 | 0.0 |
| OR-Tools-FJ (10s) | 35.66 | 6.0 |
| OR-Tools-RS (10s) | 49.75 | 9.08 |
| OR-Tools (10s) | 52.41 | 10.25 |
| OR-Tools (30s) | 53.58 | 11.16 |
| OR-Tools (60s) | **53.67** | 11.66 |
| ANYCSP (10s) | 0.00 | 0.00 |
| ConsFormer (10s) | 52.60 | **11.92** |

# H. Penalty Functions Design

As detailed in Section 3.3, our continuous penalties are designed such that

$$p(X) = 0 \iff c(X) = \texttt{True},$$

where penalty $p$ approximates constraint $c$ defined over variables $X$. Intuitively, the penalty $p$ only evaluates to 0 when the constraint $c$ is satisfied.

This approach follows constraint-based local search (Hentenryck & Michel, 2009). Here, a constraint $c$ is associated with a "violation degree" function $v_c$ where

$$v_c(X) = 0 \iff c(X) = \texttt{satisfied}$$

Specific functions to evaluate violation degrees are designed for different global constraints. For example, a violation function for $\texttt{AllDifferent}(x_1, \ldots, x_n)$ can be defined as

$$v_c(x_1, \ldots, x_n) = \sum_{i \in S} \max\left(0, \ |\{x_j = i \mid j \in 1, \ldots, n\}| - 1\right)$$

where $S$ is the set of all values in the domains of $x_1, \ldots, x_n$. Intuitively, this violation degree counts how many values are assigned to more than one variable among $x_1, \ldots, x_n$. This idea is extended to design the continuous penalty function for $\textsc{AllDifferent}_{m>n}(x_1, \ldots, x_n)$.

The design of the penalty functions is a flexible and modular component of our framework, and can benefit from further improvements which we leave for future work.

# I. Direct Gradient Descent on Variables

A natural baseline to consider is optimizing variable assignments directly using stochastic gradient descent (SGD), without a learned architecture. Specifically, if we ignore the Transformer component of ConsFormer and instead treat the variable assignments as continuous parameters, we can optimize them using our self-supervised loss. In theory, this procedure should lead to a relaxed satisfying solution.

However, we found that in practice, this method frequently converges to poor local optima. Additionally, because the optimization is performed independently for each instance, the updates cannot generalize to other instances.

To illustrate the limitation of this approach, we performed a simple experiment on Sudoku. Starting from a random initialization of missing cells, we applied SGD for 10000 steps using the self-supervised loss. The table below reports the average number of satisfied $\texttt{AllDifferent}$ constraints (out of 27 total) across 10 runs:

*Table 12.* Direct SGD optimization of Sudoku variable assignments

| # Missing Cells | 19 | 33 | 41 | 47 |
| --- | --- | --- | --- | --- |
| # Satisfied $\texttt{AllDifferent}$ | 26.8 | 25.8 | 24.5 | 21.8 |

As expected, performance degrades as the number of missing cells increases. The optimization becomes harder, and the model fails to satisfy all constraints. This highlights the importance of learning a iterative improvement model, rather than relying solely on instance-specific gradient descent.

