# OpenReview forum: "Self-Supervised Transformers as Iterative Solution Improvers for Constraint Satisfaction"
_ICML.cc/2025/Conference — ICML 2025 poster_

### Official Review · Reviewer_g9ez · 2025-03-07

**Overall Recommendation:** 2

**Summary:**

This paper mainly focuses on constraint satisfaction problems (CSPs). It proposes a transformer-based model to serve as an iterative solution improver, repeatedly revising the generated CSP solution until it is correct.

- The main _idea_ is to leverage the self-supervised learning paradigm, that is, to train the model without using any labeled or pre-processed solution.
- The main *contribution* is that this work adopts a self-supervised manner, which can avoid the pre-given solution for learning.

**Claims And Evidence:**

There are a few claims that I think may not be supported in this paper:

- 1.  The author claim that their paradigm, compared with RL-based methods, is more practical. But there is no discussion about RL-based methods both in theoretical analysis and experimental comparison.

- 2. The author compared their method with other SOTAs in table 2, however, it seems to simply copy the results heavily from Du et al. (2024). This is not fair since the results are not based on the same base model. To my best knowledge, Du et al. (2024) do this experiment based on ResNet.

**Essential References Not Discussed:**

N/A

**Experimental Designs Or Analyses:**

Yes, I have checked the experiments. The main concern of this experiment is the unfair comparison between this work and others, especially copied from Du et al. (2024). Those methods are based on a ResNet structure, while this is a transformer-based model. Also, since the authors claim their method can avoid disadvantages of RL-based method, they should discuss more about by experiments.

**Methods And Evaluation Criteria:**

The proposed model makes sense.

**Other Comments Or Suggestions:**

Overall, this paper presents an interesting approach to CSP solving and offers an effective method. However, it is not yet ready for publication due to unsupported claims and unfair comparisons. Additionally, there needs to be more discussion and comparison regarding RL-based methods.

One suggestion I have is that if the unsupported claims can be substantiated, either theoretically or practically, the paper would be significantly improved. If these concerns are adequately addressed, I would be inclined to raise my score.


## After Reviewing the Rebuttal

I appreciate the author's response; however, I find the comparison based on different architectures (backbones) to be flawed. Additionally, I disagree with the following statement:

> "We agree that a broader study—e.g., isolating the effect of ResNet vs. GNN vs. SATNet vs. Transformer under various learning paradigms—would be highly valuable, and we share your interest in such work. However, we view that as a larger, survey-level endeavor rather than the goal of the current contribution."

If the comparison is deemed unfair, it should not be included as a method of evaluation in the scope of this paper. I believe that comparing these works using the same architecture (in this case, the Transformer) would be more beneficial and equitable.

Furthermore, the authors mention a theoretical discussion of reinforcement learning (RL) methods; however, this discussion lacks formal analysis and seems to lean more towards narrative rather than substantive content, which appears to be an overstatement in their rebuttal.

Considering these issues, I believe a weak reject is an appropriate score, and I think this paper has the potential for improvement.

**Other Strengths And Weaknesses:**

PROS:
- This paper is overall well written and easy to follow.
- The proposed method makes sense, and I have run the code to validate the overall effectiveness.


CONS:
- The comparison of experiments is somehow unfair and too simple.

**Questions For Authors:**

Q1: If the iterative improvement method is effective, can this optimization approach also be applied to directly solve Constraint Satisfaction Problems (CSP) using local search? Furthermore, can iterative improvement be achieved by addressing an optimization problem with Stochastic Gradient Descent (SGD)?

Q2: Why do reinforcement learning (RL) methods not fit into this scenario? I'm confused because the reward function or signal in this setting seems much easier to define. For instance, we could simply set the reward as the difference in the number of satisfied constraints.

**Relation To Broader Scientific Literature:**

The key contributions *may* aid the development of large-scale CSP solving; however, the author did not discuss this perspective in the current version.

**Theoretical Claims:**

N/A

---

> ### Author Rebuttal · Authors · 2025-03-31
>
> We appreciate your thoughtful comments and are glad that you see the potential for our method in aiding the development of large-scale CSP solving.
>
> >no discussion about RL-based methods both in theoretical analysis and experimental comparison.
>
> We would like to clarify that both theoretical and experimental comparisons with RL-based approaches are included in the paper.
> - Theoretical considerations are discussed in Sections 2.2 and 3.3.
> - Experimentally, we directly compare against ANYCSP, a state-of-the-art RL-guided neural solver for CSPs.
>
> We will revise the paper to make these points more prominent.
>
> > The reward signal in this setting seems much easier to define. For instance, we could simply set the reward as the difference in the number of satisfied constraints.
>
> Defining a reward as the number of satisfied constraints is indeed an intuitive design. However, prior work indicates that such rewards are not sufficient to guide learning effectively.
>
> ANYCSP [1] experimented with the exact reward you have suggested and found that “models trained with this reward tend to get stuck in local maxima”. They comment that “intuitively, this simple reward scheme immediately punishes the policy for stepping out of a local maximum causing stagnating behavior”.
>
> Their method implements a much sparser reward to avoid this issue by setting the reward to 0 in any step in which the new assignment is not an improvement over the previous best.
>
> In contrast, our approach provides a dense training signal through differentiable penalty functions. These loss functions guide the model even when no constraint is fully satisfied.
>
> [1] Tönshoff, Jan, et al. "One model, any CSP: graph neural networks as fast global search heuristics for constraint satisfaction." (2022).
>
>
> >Comparison to other methods is not fair. Du et al. (2024) do this experiment based on ResNet.
>
> We note that Du et al. themselves compare against models like SATNet and RRN which are not ResNet based models. SATNet is a model that learns the SDP relaxation of a SAT formulation, while RRN is a graph-based model.
>
> While it’s true that Du et al.’s method uses a ResNet base model, We follow standard practice  and make comparisons between full frameworks for solving CSPs.
>
>
> > can this optimization approach also be applied to directly solve Constraint Satisfaction Problems (CSP) using local search?
>
> Yes! And indeed the existing literature of local search was a prominent inspiration for us, specifically, our approach is grounded in principles from the constraint-based local search literature [2]. In this approach, constraints are associated with “violation degrees” that measure how far a solution is from satisfying the constraint (violation = 0 ⇔ satisfied). We adopt this principle and extend it to the neural domain by designing differentiable approximations of these violations that work with probabilistic assignments.
>
> [2] Michel, L., Hentenryck, P.V. (2018). Constraint-Based Local Search. In: Martí, R., Pardalos, P., Resende, M. (eds) Handbook of Heuristics.
>
> >Furthermore, can iterative improvement be achieved by addressing an optimization problem with Stochastic Gradient Descent (SGD)?
>
> This is a great insight. Indeed if we ignore the transformer architecture and perform SGD on a set of variable assignments guided by our self-supervised loss, the variable assignments can be updated to become a (relaxed) satisfying solution. However, we found that in practice this method often leads to local optima, furthermore, we see a drastic decrease in performance as the SGD focus on a single instance and does not know how to generalize to other instances. Our framework essentially learns this update using a single-step transformer and is able to generalize by training on a larger set of instances.
>
> To showcase this, we run a simple SGD on a Sudoku board where the missing values are randomly initialized and update it with SGD guided by the loss function for 10000 iterations. We observe the following results averaged over 10 runs:
> |#Missing Values|19|33|41|47|
> |-|-|-|-|-|
> |#AllDifferent Satisfied (Max 27)|26.8|25.8|24.5|21.8|
>
> We see as the problem becomes harder with more missing cells, the number of satisfied alldifferent constraints became lower. We will include this discussion in the revision of the paper.

---

> > ### Comment · Reviewer_g9ez · 2025-04-02
> >
> > Thank you for your efforts.
> >
> > After reviewing your response, I have some concerns:
> >
> > - The authors state that "theoretical considerations are discussed in Sections 2.2 and 3.3." However, I did not find any theoretical considerations in these sections. Section 2.2 focuses on related work, while Section 3.3 describes the loss function.
> >
> > - My main concern is that the authors suggest they are using different backbones for various comparison methods. While I acknowledge that there may be mistakes in previous work, I encourage this study to address and rectify this issue.

---

> > > ### Author Response · Authors · 2025-04-05
> > >
> > > Thank you for your continued engagement with our work. We appreciate the opportunity to clarify your remaining concerns.
> > >
> > > > Theoretical considerations not found. Section 2.2 focuses on related work, while Section 3.3 describes the loss function.
> > >
> > > Apologies for the confusion, Section 2.2 introduces RL approaches from the literature, and Section 3.3 discusses the theoretical disadvantages of training with RL compared to training with self-supervision. We appreciate the opportunity to elaborate further and make these theoretical motivations more explicit in the paper.
> > >
> > > Our key point is the following: while RL is appropriate for sequential decision-making with a **black-box reward function** provided by an environment, the reward function for a CSP is not a black box. Specifically, if we view the degree of constraint satisfaction as the reward, then this is a “white box” for which we know the analytical form (i.e., the expression representing a constraint). What remains is to make the degree of constraint satisfaction (or the negative of the degree of constraint violation) differentiable w.r.t. an assignment of values to variables. This enables end-to-end gradient descent from an input (initial assignment) to an output (improved assignment) to a loss function (degree of constraint violation).
> > >
> > > Any RL-based method for CSP must squash the extremely rich signal of constraint violation into a single scalar reward, aggregated over the multiple constraints. Gradients do not flow directly from this reward to the assignment. Any performant RL policy must “learn” the relationship between the action (i.e., an assignment) and the reward (i.e., degree of constraint satisfaction) in order to maximize the reward. This is unnecessarily complicated for a fully observed CSP.
> > >
> > > For example, as discussed previously, ANYCSP [1] implements a very sparse reward function, setting the reward to zero unless a strictly better assignment satisfying more constraints is found. In contrast, our self-supervised formulation introduces continuous penalty functions that provide dense, gradient-based supervision, even for partially incorrect assignments.
> > >
> > > Practically, this translates to **significantly faster training**. While ANYCSP reports 24–48 hours for full training, our model achieves strong results in under 6 hours, and fully converges within 10 hours.
> > >
> > > [1] Tönshoff, Jan, et al. "One model, any CSP: graph neural networks as fast global search heuristics for constraint satisfaction." (2022).
> > >
> > > > Backbones differ across methods; this needs to be addressed or rectified.
> > >
> > > We appreciate this concern and agree that comparing different architectures under different paradigms can be problematic if the goal is to isolate performance of a specific training paradigm. However, that is not the goal of our paper.
> > >
> > > The goal of our paper is to introduce a new heuristic solving framework: a Transformer architecture trained via self-supervised learning, designed specifically for CSPs. Our comparisons are therefore made against other end-to-end neural/traditional CSP solvers, each using their own architectural and learning choices. This is in line with the evaluation practices of all related work.
> > >
> > > We are not using different backbones while comparing different methods, but comparing our method (Transformer + self-supervised learning) against other complete solver pipelines.
> > >
> > > That said, our experiments showed significantly better performance than Yang et al. [2], who use a Transformer backbone similar to ours—making this a particularly direct and meaningful comparison.
> > >
> > > We agree that a broader study—e.g., isolating the effect of ResNet vs. GNN vs. SATNet vs. Transformer under various learning paradigms—would be highly valuable, and we share your interest in such work. However, we view that as a larger, survey-level endeavor, rather than the goal of the current contribution.
> > >
> > > [2] Yang, et. al. "Learning to Solve Constraint Satisfaction Problems with Recurrent Transformer." The Eleventh International Conference on Learning Representations.

---

### Official Review · Reviewer_3qja · 2025-03-09

**Overall Recommendation:** 2

**Summary:**

This paper presents a Transformer-based learning framework for solving CSPs. Specifically, they leverage a transformer architecture to refine the solution, where they show that decision variable position encoding is key for transformer learning. They adopt a continuous approximation as loss function, and show on three CSPs (Sudoku, Graph Coloring, Nurse Scheduling) that their proposed method outperform the baselines.

**Claims And Evidence:**

The claims made in the submission generally are supported by clear and convincing evidence, although I have some questions / concerns regarding the evaluation. See my answer to “Experimental Designs or Analyses”.

**Essential References Not Discussed:**

N/A

**Experimental Designs Or Analyses:**

I see the following issues regarding experimental design:
- The authors set a time limit of 10s for solving graph coloring and nurse scheduling. I find the time limit to be too short (e.g. maybe this reflect that the CSPs the authors evaluation on are too simple to solve). Can the authors benchmark on more complicated problems? Also, can the authors increase the time limit for solving the instances? I think it makes sense for constraint programming solver CP-SAT to take more time than learning-based methods, so the authors should consider allowing CP-SAT to solve for longer.
- Can the authors compare with more heuristic baselines (e.g. local search, genetic algorithm)? I find the comparison with heuristic baselines somewhat lacking.
- Maybe I missed this, but I don’t see nurse scheduling results reported in the paper?

**Methods And Evaluation Criteria:**

The proposed methods and evaluation datasets seem to be reasonable, except for my concerns in “Experimental Designs or Analyses”.

**Other Comments Or Suggestions:**

Sec 2.2. “Yang et al. proposed a recurrent Transformer architecture” → You missed the year citation for Yang et al.

**Other Strengths And Weaknesses:**

I’m concerned that the paper may lack novelty: for example, encodeing positional information for iterative solution improvement has been explored in learning for combinatorial optimization literature before, and the Gumble-softmax differentiable loss isn’t new either.

**Questions For Authors:**

1. Sec 3.3 Eq (5): “hyperparameters lambda_i is the weight assigned to p_i”. How did you decide the lambda_i? Would an adaptive way to select different lambda_i l for different constraints lead to better performance? (potentially with learning, or some heuristic methods)
2. Instead of the differentiable loss with GumbelSoftmax, I wonder if the authors have tried reinforcement learning? The benchmark tasks seem to be relatively standard to design RL rewards, so I feel like an ablation study of differentiable loss v.s. RL would further strengthen the paper.
3. Sec 3.4 Iterative Test-Time Deployment: instead of feeding the entire solution back for refinement, have the authors consider only improving a subpart of the solution and fixing the rest (e.g. incorporate a local search procedure with learning)? How well would that work?

**Relation To Broader Scientific Literature:**

This paper contributes an improved learning methods for solving constraint programming problems. I find the discussions / experiments regarding position encoding interesting.

**Theoretical Claims:**

There’s no theoretical claims nor proofs in this paper.

---

> ### Author Rebuttal · Authors · 2025-03-31
>
> Thank you for the thoughtful feedback and questions. We’ve conducted new experiments and clarified key aspects of the method in response to your concerns, which we believe strengthen the submission significantly.
>
> > Can the authors benchmark on more complicated problems?
>
> We have added experiments on the MAXCUT problem. Please refer to our response to reviewer tcHx for details.
>
> > CP-SAT takes more time than learning-based methods, so the authors should consider allowing CP-SAT to solve for longer.
>
> Efficiency is indeed a major advantage of our method and in many applications it is necessary to solve for a good feasible solution very quickly.
>
> That said, we conducted new experiments where CP-SAT was allowed to run for 30 and 60 seconds on the Graph-Coloring task with 10 colors (where ConsFormer had previously outperformed it under 10s). Results are shown below:
> ||n=100|n=200|
> |-|-|-|
> |OR-Tools(10s)|52.41|10.25|
> |OR-Tools(30s)|53.58|11.16|
> |OR-Tools(60s)|53.67|11.66|
> |ConsFormer(10s)|52.60|11.92|
>
> We see that CP-SAT can outperform our method on small instances (nodes=100) with extended time, but it still underperforms on larger instances (nodes=200), even with 6x more time. Furthermore, in the new MAXCUT problem, 20 parallel runs–each with 180s limit–were used to compute the results.  ConsFormer also outperforms OR-Tools by a significant margin.
>
> > compare with more heuristic baselines
>
> We include 3 additional heuristic baselines for graph coloring:
> - Greedy Coloring
> - Feasibility Jump
> - Random Search
>
> The first is the greedy coloring algorithm implemented by networkx and the other two are local search approaches implemented by OR-Tools. We note that the LS algorithms are already present in the default OR-Tools used for the paper.
> |Color count = 10|n=100|n=200|
> |-|-|-|
> |Greedy|0.75|0.00|
> |FJ(10s)|35.66|6.0|
> |RS(10s)|49.75|9.08|
> |OR-Tools(10s)|52.41|10.25|
> |ANYCSP(10s)|0|0|
> |ConsFormer(10s)|52.60|11.92|
>
> |Color count = 5|n=50|n=100|
> |-|-|-|
> |Greedy|32.42|0.00|
> |FJ(10s)|82.83|54.5|
> |RS(10s)|83.08|56.91|
> |OR-Tools(10s)|83.08|57.16|
> |ANYCSP(10s)|79.17|34.83|
> |ConsFormer(10s)|78.16|42.50|
>
> We observe that while the local-search based heuristics were able to perform well on the smaller instances, their performance significantly worsens on the larger instances with 10 colors.
>
> > nurse scheduling results
>
> They were reported in-text in Section 4.3. We will revise the paper for better clarity.
>
> > I’m concerned that the paper may lack novelty.
>
> As far as we know, we are the first to successfully apply a Transformer architecture to solve general CSPs in a self-supervised setting. Existing work often requires graph representations of the problems while we treat them as sequences.
>
> > How did you decide the lambda_i? Would an adaptive way to select different lambda_i l for different constraints lead to better performance?
>
> This is a great suggestion and an adaptive lambda is indeed an interesting direction that we plan on exploring for future work. For now, each problem has a relatively simple mixture of constraints, so the weights are manually set.
>
> >Instead of the differentiable loss with GumbelSoftmax, I wonder if the authors have tried reinforcement learning?
>
> We apologize if we have misunderstood, but we believe the reviewer may be referring to use of the REINFORCE trick (or score-function based gradient estimators) from RL that can be generally used to compute gradients of discrete variables.  However, we note that the Gumbel-Softmax paper[1] discusses that this gradient estimator has high variance (Section 3.2 of [1]) and performs very poorly (Section 4.1 of [1]) in practice -- in fact it is the worst performing of all discrete gradient estimators they compare.
>
> [1] Jang et al. "Categorical Reparameterization with Gumbel-Softmax."
>
> >instead of feeding the entire solution back for refinement, have the authors consider only improving a subpart of the solution and fixing the rest
>
> We completely agree with the intuition, and in fact, our model effectively incorporates this idea via the Variable Subset Selection step (Section 3.2, line 203). At each iteration, only a subset of variables is selected for update, mimicking local search behavior. We found that this strategy significantly improves generalization compared to updating the full assignment each time, as discussed in Section 4.4.

---

### Official Review · Reviewer_QdQE · 2025-03-14

**Overall Recommendation:** 5

**Summary:**

The paper introduces an iterative, local search approach towards solving constraint satisfaction problems with transformer models and self-supervision.
A new encoding scheme and the self-supervised training scheme are introduced plus a differentiable approximation for the violation of key constraints. The main aspect is the combination of being able to differentiate the approximate constraints and the 1-step prediction approach of the transformer with an overall iterative solving loop.
Experiments are performed and show the effectiveness of the approach.

## update after rebuttal
I thank the authors for their response and their effort during the rebuttal. I maintain my score.

**Claims And Evidence:**

The paper claims to avoid bottlenecks available in other approaches (supervised + RL) and does so through the self-supervision approach.
This is not novel (RL does it somehow), but it's more efficient because there is differentiation.

It is more limited though, because it needs support for every constraint to be differentiable. Seeing the approximations proposed in the paper, these differentations are a conceptualization of the constraint violations, which are commonly used in constraint-based local search too to judge the quality of an improvement, and should be derivable for every constraint. So, at this stage it's mostly a matter of defining them for all constraints and, maybe, there can even be a more general form derived by looking the overall CSP. In conclusion, the claim is sufficiently supported.

**Essential References Not Discussed:**

I'm not aware of any missing literature.

**Experimental Designs Or Analyses:**

Experiments make sense. Baselines are okay. There is a lot of work in this area and other baselines might be suitable as well besides ANYCSP, but including OR-Tools is the most relevant one and they are sufficient to serve the point.

**Methods And Evaluation Criteria:**

The selected methods make sense. A CSP is inherently either a graph or a sequence and using a transformer model is one way to model it in the ML paradigm (graph-based approaches being the other major one currently studied in the literature).

In terms of datasets, the selection is okay and in line with the literature. In CSP practice there are more complex and larger datasets available, which should be considered for future work and would really put a test on the scalability of the approach -- but not necessarily required here.

**Other Comments Or Suggestions:**

- The results for nurse scheduling were a bit lost in the text, maybe you can include them in Table 3

**Other Strengths And Weaknesses:**

This is a really good paper with a strong contribution. The proposed encoding and training scheme is sufficiently simple to be usable and adaptable for other settings and by relying on CP modelling for the CSP problems it can be very powerful.

**Questions For Authors:**

- Have you evaluated the performance when decomposing the global constraints? How does the model balance having more variables vs more complex global constraints to learn?
- Can you transfer learn, i.e. pretrain the model for individual constraints first and then merge them towards more complex problems?

**Relation To Broader Scientific Literature:**

Connections to the relevant areas (CP, ML, ML4CO) are made and supported with relevant references.

**Theoretical Claims:**

N/A

---

> ### Author Rebuttal · Authors · 2025-03-31
>
> We sincerely appreciate the encouraging feedback and thoughtful comments.
>
> > self-supervision is more limited because it needs support for every constraint to be differentiable.
>
> You're absolutely right to point out that a key limitation of the self-supervised approach is the need to define differentiable approximations for each constraint. As you noted, our method draws directly from the constraint-based local search literature [1], where constraint violations are quantified using "violation degrees" to guide iterative improvement. Our contribution builds on this idea by adapting it into a differentiable, neural-friendly form that allows gradient-based learning.
>
> [1] Michel, L., Hentenryck, P.V. (2018). Constraint-Based Local Search. In: Martí, R., Pardalos, P., Resende, M. (eds) Handbook of Heuristics.
>
> >In CSP practice there are more complex and larger datasets available, put a test on the scalability of the approach.
>
> This is a valuable point and was also raised by other reviewers. In response, we have added new experiments on the MAXCUT problem (see our response to Reviewer tcHx). These instances include thousands of variables and constraints and serve as a stronger test of scalability. The results indicate that ConsFormer remains competitive, even under larger problem sizes. While our ultimate goal is to apply ConsFormer to a wide range of CSPs, we believe that the experimental results on four benchmark problems with different combinatorial structures provide ample evidence for the promise of our method.
>
> >The results for nurse scheduling were a bit lost in the text
>
> Thank you for raising this, we will revise the paper to highlight the results better.
>
> >Have you evaluated the performance when decomposing the global constraints? How does the model balance having more variables vs more complex global constraints to learn?
>
> This is an excellent question. To investigate, we conducted an additional experiment on Sudoku where the all-different constraints are decomposed to pairwise inequalities (36 inequality constraints per all-different). This can be effectively achieved by defining the continuous penalty as follows:
> $$AllDifferent(x_1, \ldots, x_n) \rightarrow \sum_{i\neq j, i< j} \left( \sum_{k} x_i^{(k)}\cdot x_j^{(k)} \right)$$
> We present the instance solved percentage below:
>
> |#Iterations|AllDifferent|Decomposed|
> |-|-|-|
> |1K|59.22|58.47|
> |2K|65.88|65.91|
> |10K|77.74|77.96|
>
> Interestingly, we observe that in this case, decomposing the constraints does not significantly affect performance which suggests that the model can equally handle a larger number of simpler constraints. However, we note that the resulting number of constraints is still relatively small after decomposing alldifferent for sudoku. Other more complex problems with global constraints may reveal different trade-offs which we leave for future work.
>
> >Can you transfer learn, i.e. pretrain the model for individual constraints first and then merge them towards more complex problems?
>
> This is a great insight and is something we are planning as a future direction! One limitation of our current approach (as well as other neural approaches) is the need to train a new model for a new problem domain that consists of a different combination of constraints. With the success of the transformer’s pre-train + fine-tune approach in the natural language setting [2], one could imagine this approach applied to our setting, where models are first pre-trained for all constraints, then combined modularly to be fine-tuned for new problems. Despite the potential promise of this direction, there are many open questions to resolve in future work in order to bridge from the current model to this ideal goal of a highly general and transferable pretrained model.  We will add this excellent future work discussion on revision.
>
> [2] Radford, Alec, et al. "Improving language understanding by generative pre-training." (2018).

---

> > ### Comment · Reviewer_QdQE · 2025-04-02
> >
> > Thank you for your response to my comments and for providing additional results!
> >
> > I remain convinced that this is a great contribution and maintain my score.

---

### Official Review · Reviewer_tcHx · 2025-03-15

**Overall Recommendation:** 3

**Summary:**

The paper proposes training a transformer architecture in a self-supervised manner to solve constraint satisfaction problems. The input to the model is an assignment of values to the variables and the output is a refined assignment. The assignment is encoded as tokens and the constraints of the problem are encoded as relative positional encodings. The loss used to train the model is a continuous proxy for constraint satisfaction. Each type of constraint has its own proxy and then a weighted combination is taken to compute the final loss.

The approach shows competitive results in sudoku solving and CSPs like graph coloring and the nurse scheduling problem.

**Claims And Evidence:**

See my other comments.

**Essential References Not Discussed:**

Below I provide a non-exhaustive list of references that should be mentioned in the context of constraint satisfaction.

For CSPs and constraint satisfaction:
- (self-supervised architecture for CSPs) Toenshoff, Jan, et al. "Graph neural networks for maximum constraint satisfaction." Frontiers in artificial intelligence 3 (2021): 580607.
- (cardinality constraints) Wang, Runzhong, et al. "LinSATNet: the positive linear satisfiability neural networks." International Conference on Machine Learning. PMLR, 2023.
 - (on using gnns for CSPs) Yau, Morris, et al. "Are Graph Neural Networks Optimal Approximation Algorithms?." Advances in Neural Information Processing Systems 37 (2024): 73124-73181.

For SAT:
- (Self-supervised sat solving) Ozolins, Emils, et al. "Goal-aware neural SAT solver." 2022 International joint conference on neural networks (IJCNN). IEEE, 2022.
-  (RL for satisfiability without labels)Yolcu, Emre, and Barnabás Póczos. "Learning local search heuristics for boolean satisfiability." Advances in Neural Information Processing Systems 32 (2019).

For loss function design:
-  (continuous relaxation/extension design) Karalias, Nikolaos, et al. "Neural set function extensions: Learning with discrete functions in high dimensions." Advances in Neural Information Processing Systems 35 (2022): 15338-15352.
 - (for combinatorial constraints in self-supervised learning )Bu, Fanchen, et al. "Tackling prevalent conditions in unsupervised combinatorial optimization: Cardinality, minimum, covering, and more." arXiv preprint arXiv:2405.08424 (2024).
-  (loss function design for self-supervised constrained optimization) Karalias, Nikolaos, and Andreas Loukas. "Erdos goes neural: an unsupervised learning framework for combinatorial optimization on graphs." Advances in Neural Information Processing Systems 33 (2020): 6659-6672.

**Experimental Designs Or Analyses:**

While the choice of benchmarks in the paper is sensible, I think the overall experimental evaluation and the ablations are lacking for a paper whose content is almost exclusively empirical. Below I provide a list of issues that I have with the evaluation in the paper:

-  The paper compares the proposed transformer approach against Anycsp on particular problems like k-coloring but does not provide some of the other comparisons that could be done with Anycsp. For instance, Anycsp has really strong results on max-cut and also performs pretty well on SAT so seeing how this transformer approach stacks up on those benchmarks would be nice. Another candidate benchmark can be found in [1]. These are CNF-sat instances coming from different combinatorial problems.

- Another comment related to the point above has to do with the scalability of the approach. Most of the results provided are evaluations on smaller instances. Coming back to the max-cut example from Anycsp, there are instances there with several thousands of edges (constraints). Can the proposed transformer approach scale up to that? If not, what are the current limitations and are there ways that they could be addressed?

1. Li, Zhaoyu, Jinpei Guo, and Xujie Si. "G4SATBench: Benchmarking and advancing sat solving with graph neural networks." arXiv preprint arXiv:2309.16941 (2023).

**Methods And Evaluation Criteria:**

The evaluation makes sense and the CSP problems that were picked for the evaluation are reasonable.

**Other Comments Or Suggestions:**

N/A

**Other Strengths And Weaknesses:**

- It is unclear to me what the purpose of the Gumbel-Softmax approach is when predicting the value for the variables. Why is the gumbel noise added? Isn't the softmax sufficient? Does the randomness somehow help? If yes, an ablation study is appropriate. The choice is not discussed in the paper so it's hard to figure out what it could achieve.

Overall, I find this direction promising but the experimental evaluation is inadequate so I cannot recommend accepting this. The writing could also be improved by providing a more detailed discussion around the questions I have raised in this review.

**Questions For Authors:**

- There has been work on encoding cardinality cardinality constraints (and other constraints) into the loss functions of neural networks [1]. The specific choice of functions that are used as proxies for constraint violation in the loss seems ad-hoc. Have the authors compared with other techniques from the literature? For example there are various published works on loss function design for self-supervised constrained optimization [1,2].  Have the authors compared those techniques to the ones proposed here?  Using a more ad-hoc approach is fine it works but I think it merits more discussion in the paper because there are different ways of encoding those constraints and their performance may vary.

1. Bu, Fanchen, et al. "Tackling prevalent conditions in unsupervised combinatorial optimization: Cardinality, minimum, covering, and more." arXiv preprint arXiv:2405.08424 (2024).
2. Karalias, Nikolaos, and Andreas Loukas. "Erdos goes neural: an unsupervised learning framework for combinatorial optimization on graphs." Advances in Neural Information Processing Systems 33 (2020): 6659-6672.

**Relation To Broader Scientific Literature:**

The paper aims to provide a new self-supervised pipeline for constraint satisfaction problems. As I mention in other places in this rebuttal this has been explored with other architectures before. The loss function design for constraint satisfaction aspect has also been explored in the literature in the self-supervised setting.

The main innovation here is to create a neural solver that relies on a powerful architecture like the transformer. This is obviously worth exploring since transformers have been extremely successful and so far there hasn't been a particularly compelling transformer architecture for these kinds of problems in the literature.

**Theoretical Claims:**

N/A

---

> ### Author Rebuttal · Authors · 2025-03-31
>
> We greatly appreciate your thorough feedback and have conducted new experiments which we believe have substantially improved the paper.
>
> > Additional benchmark.
>
> We have adapted ConsFormer for MAXCUT based on your suggestion.
> - MAXCUT is the problem of partitioning nodes of a graph into two sets in a way that maximizes the size of the cut.
> - Following ANYCSP, we train on graphs with 100 vertices and test on GSET instances with 800 to up to 10000 vertices.
>
> We report the absolute and relative gap (in percentage) to best known values:
> |Method|V=800|V=1K|V=2K|V≥3K|
> |-|-|-|-|-|
> |Greedy|411.44(5.26)|359.11(6.64)|737.00(6.81)|774.25(6.30)|
> |SDP|245.44(3.14)|229.22(4.24)|-|-|
> |RUNCSP|185.89(2.38)|156.56(2.90)|357.33(3.30)|401.00(3.26)|
> |ECO-DQN|65.11(0.83)|54.67(1.01)|157.00(1.45)|428.25(3.49)|
> |ECORD|8.67(0.11)|8.78(0.16)|39.22(0.36)|187.75(1.53)|
> |ANYCSP|1.22(0.02)|2.44(0.05)|13.11(0.12)|51.63(0.42)|
> |ConsFormer|24.44(0.31)|18.22(0.34)|47.00(0.43)|155.88(1.27)|
> |OR-Tools|143.89(1.84)|112.78(2.09)|365.89(3.38)|378.62(3.08)|
>
> We observe that while ANYCSP remains the best performing approach on GSET, ConsFormer achieves a relative gap of 0.31% to 1.27% on average without extensive model tuning, showcasing its ability to scale up to larger problems with thousands of constraints.
>
> > ablation study on the Gumbel-Softmax.
>
> We chose Gumbel-Softmax for its differentiable sampling of discrete variables, aligning with the discrete nature of CSPs. Experiments during development showed it outperformed standard softmax, which we confirmed through a more systematic ablation below.
>
> However, your question prompted us to further investigate why Gumbel-Softmax helps: is it the stochasticity introduced by the Gumbel noise, or simply the sharper outputs it produces? To investigate this, we ran additional experiments using softmax with added temperature control – a variant we had not systematically explored in earlier versions ($Softmax_\tau(z_i) = \frac{\exp\left(z_i / \tau\right)}{\sum_{j} \exp\left(z_j / \tau\right)}$
> ), and present the results below (for Sudoku/Graph Coloring, we show the % instances solved, MAXCUT reports the absolute gap to best known values):
>
> ||Gumbel-Softmax|Softmax|Softmax w/T|
> |-|-|-|-|
> |Sudoku|100|100|100|
> |Sudoku-hard|77.74|73.72|**85.67**|
> |Graph-Coloring-5 V=50|**78.16**|74.91|77.33|
> |Graph-Coloring-5 V=100|42.50|35.33|**42.66**|
> |Graph-Coloring-10 V=100|52.60|53.00|**53.66**|
> |Graph-Coloring-10 V=200|11.92|12.75|**12.92**|
> |MAXCUT V=800|**24.44**|126.56|123.11|
> |MAXCUT V=1K|**18.22**|56.89|58.33|
> |MAXCUT V=2K|**47.00**|135.11|123.67|
> |MAXCUT V≥3K|**155.88**|305.25|287.38|
>
> Interestingly, softmax with temperature yielded similar or better results than Gumbel-Softmax on problems with smaller sizes, while performing worse on larger problems from MAXCUT. This suggests that while the sharper distributions indeed boost model performance, the stochasticity allows the model to generalize better across larger problem instances. The randomness can promote diversity in intermediate solutions which may help the model escape local optima over multiple inference steps. We will include this ablation in the updated paper.
>
> > additional references
>
> We appreciate the extensive list of references and will include them accordingly. We note that this list of references echoes our own finding which is that most existing approaches focus on graph-based representations or SAT formulations. To the best of our knowledge, Transformers have not been effectively implemented for general form CSPs.
>
> > constraint violation functions seems ad-hoc and merits more discussion in the paper.
>
> Our approach is grounded in principles from the constraint-based local search literature in CP [1]. Here, constraints are associated with “violation degrees” where violation degree = 0 ⇔ satisfied. Specific functions to evaluate violation degrees are designed for different global constraints.
>
> Selecting the design for the continuous penalty is indeed important. During development we experimented with various continuous approximations including some inspired by recent work using T-norm [2] and Entropy [3] with varying effectiveness.
>
> However, our goal is to showcase the effectiveness of the self-supervised approach combined with the single-step transformer. We view the choice of continuous penalty functions as a flexible and modular component of our framework, one that can benefit from further improvements, but is not the central focus of this work. We will include more discussion on this in the updated paper.
>
> [1] Michel, L., Hentenryck, P.V. (2018). Constraint-Based Local Search. In: Martí, R., Pardalos, P., Resende, M. (eds) Handbook of Heuristics.
>
> [2] Giannini, Francesco, et al. “T-norms driven loss functions for machine learning.”
>
> [3] Chen, Di, et al. “Deep reasoning networks: Thinking fast and slow.”

---

> > ### Comment · Reviewer_tcHx · 2025-04-07
> >
> > OK, thank you for the update. I will raise my score but I am still not completely satisfied with the experiments here. The GSET results look promising. I would like to see how long it takes to train/test for those. You mention that it takes 6-10 hours to train for your original experiments. From your response to the other reviewer, it sounds like it's a similar amount of time. What about memory? I think showing more results for the CNF instances I brought up and doing a study on size generalization would make the case for this paper stronger.
> >
> > By size generalization, I mean: suppose you pick hard 3-CNF random sat instances around the critical threshold (~4.25). How large are the instances that you can solve? This includes discussing the memory cost/scalabity of the approach, the scale X of instances of the training set needed (also how many) to do well at a scale of Y. Scale as measured in number of clauses or variables.
> >
> >  I will bump my score up because the paper shows the potential of training a transformer architecture in self-supervised style for combinatorial problems, but I think those kinds of experiments are essential for papers with an empirical focus. I am willing to propose acceptance for this as long as the authors commit to providing more of the results I suggested (say in the final version of the paper if not possible soon).

---

> > > ### Author Response · Authors · 2025-04-08
> > >
> > > Thank you for your continued engagement and suggestions.
> > >
> > > > I would like to see how long it takes to train/test for those.
> > >
> > > Due to time constraints of the rebuttal, we ran limited hyper-parameter search and used relatively smaller models. The reported results are based on a model trained for under three hours (wall clock) with 3 attention heads and 4 transformer layers. The testing was conducted following the same procedure and time limits as ANYCSP for GSET.
> > >
> > > > What about memory?
> > >
> > > The models were trained on the same type of single-GPU nodes, each with 32 GB of CPU memory and between 12 GB and 32 GB of GPU memory, depending on the allocated GPU.
> > >
> > > > I think showing more results for the CNF instances I brought up and doing a study on size generalization would make the case for this paper stronger.
> > >
> > > We agree that a study on size generalization would be insightful and we will conduct the suggested experiments for the final version. Specifically, we plan to do the following:
> > > - Adapt our framework to 3-SAT using one-hot encoded binary variables
> > > - Explore differentiable loss functions for SAT such as the log loss used by QuerySAT[1] and various T-Norm driven loss functions (Gödel, Łukasiewicz, Product, etc)[2].
> > > - Report the 3-SAT performance in comparison to ANYCSP.
> > > - Conduct size generalization study as suggested by the reviewer.
> > >
> > > [1] Ozolins, Emils, et al. "Goal-aware neural SAT solver." 2022 International joint conference on neural networks (IJCNN). IEEE, 2022.
> > >
> > > [2] Giannini, Francesco, et al. "T-norms driven loss functions for machine learning." Applied Intelligence 53.15 (2023): 18775-18789.

---

### Decision · Program_Chairs · 2025-05-01

**Decision:**

Accept (poster)

**Comment:**

This paper proposes ConsFormer, a self-supervised Transformer framework for solving CSPs through iterative refinement. It avoids reliance on labeled solutions or complex RL rewards by using differentiable proxies for constraint violations. The idea is well-motivated and the method performs competitively across several CSP benchmarks. While some reviewers raised concerns about experimental depth and comparison fairness, the authors responded with new results (e.g., on MAXCUT) and detailed clarifications, which strengthened the case for the paper. Overall, this is a solid contribution with practical value and promising scalability.